# Overexpression of a *Fragaria vesca* MYB Transcription Factor Gene (*FvMYB82*) Increases Salt and Cold Tolerance in *Arabidopsis thaliana*

**DOI:** 10.3390/ijms231810538

**Published:** 2022-09-11

**Authors:** Wenhui Li, Jiliang Zhong, Lihua Zhang, Yu Wang, Penghui Song, Wanda Liu, Xingguo Li, Deguo Han

**Affiliations:** 1Key Laboratory of Biology and Genetic Improvement of Horticultural Crops (Northeast Region), Ministry of Agriculture and Rural Affairs, National-Local Joint Engineering Research Center for Development and Utilization of Small Fruits in Cold Regions, College of Horticulture & Landscape Architecture, Northeast Agricultural University, Harbin 150030, China; 2Horticulture Branch of Heilongjiang Academy of Agricultural Sciences, Harbin 150040, China; 3Institute of Rural Revitalization Science and Technology, Heilongjiang Academy of Agricultural Sciences, Harbin 150028, China

**Keywords:** *Fragaria vesca*, *FvMYB82*, abiotic stress, salt stress, low-temperature stress

## Abstract

The MYB transcription factor (TF) family is one of the largest transcription families in plants, which is widely involved in the responses to different abiotic stresses, such as salt, cold, and drought. In the present study, a new MYB TF gene was cloned from *Fragaria vesca* (a diploid strawberry) and named *FvMYB82*. The open reading frame (ORF) of *FvMYB82* was found to be 960 bp, encoding 319 amino acids. Sequence alignment results and predictions of the protein structure indicated that the *FvMYB82* contained the conserved R2R3-MYB domain. Subcellular localization analysis showed that *FvMYB82* was localized onto the nucleus. Furthermore, the qPCR showed that the expression level of *FvMYB82* was higher in new leaves and roots than in mature leaves and stems. When dealing with different stresses, the expression level of *FvMYB82* in *F. vesca* seedlings changed markedly, especially for salt and cold stress. When *FvMYB82* was introduced into *Arabidopsis thaliana*, the tolerances to salt and cold stress of *FvMYB82*-OE *A. thaliana* were greatly improved. When dealt with salt and cold treatments, compared with wild-type and unloaded line (UL) *A. thaliana*, the transgenic lines had higher contents of proline and chlorophyll, as well as higher activities of superoxide dismutase (SOD), peroxidase (POD), and catalase (CAT). However, the transgenic *A. thaliana* had lower level of malondialdehyde (MDA) and electrolytic leakage (EL) than wild-type and UL *A. thaliana* under salt and cold stress. Meanwhile, *Fv**MYB82* can also regulate the expression of downstream genes associated with salt stress (*AtSnRK2.4*, *AtSnRK2.6*, *AtKUP6*, and *AtNCED3*) and cold stress (*AtCBF1*, *AtCBF2*, *AtCOR15a*, and *AtCOR78*). Therefore, these results indicated that *FvMYB82* probably plays an important role in the response to salt and cold stresses in *A. thaliana* by regulating downstream related genes.

## 1. Introduction

Many factors in the growth environment of plants are changeable and may cause biotic or abiotic stress, which will hinder the growth of plants and can even cause irreversible damage [1]. In recent years, with technological advances in molecular biology, it is increasingly feasible to improve the adaptability of plants to the external environment through genetic manipulation. Transcription factors and their regulatory functions have an important place in current plant research. Transcription factors in plants are activated by specific signal transduction pathways in response to a specific stress stimulus. Transcription factors bind to the corresponding cis-acting element when activated under stress, thus activating the expression of downstream stress-related genes and thus enhancing the stress tolerance of the plant [2].

The MYB transcription factors comprise a large and multifunctional superfamily [3] and are expressed in all eukaryotes [4]. The superfamily plays an indispensable role in the metabolism and growth of plants [5]. The MYB transcription factors are classified into different families based on the number of DNA-binding domains [6], such as 1R-MYB, R2R3-MYB, 3R-MYB, and 4R-MYB2 [7]. The 1R-MYB family is involved in morphogenesis, secondary metabolism, and fruit development. The R2R3-MYB family plays a great role in response to plant abiotic stress, such as drought, low temperature, and high salt, and the 3R-MYB family regulates the cell cycle [8]. MYB transcription factors are relatively conserved, both functionally and structurally, in plants compared with other organisms. MYB proteins are abundant in plants and perform a unique function in many plant species, and they are especially characterized by roles in primary and secondary metabolic responses, as well as responses to biotic and abiotic stresses [9].

Salt stress and low temperature are common abiotic stress factors throughout the life cycle of plants. Salt stress and cold stress cause plant foliar diminution and withering, reduction in plant height, enhanced permeability of plant cell membranes, solute leakage, disruption of the metabolic system and, ultimately, growth and yield are severely affected [10,11]. In *A. thaliana*, more than 100 MYB proteins have been identified in the R2R3-MYB family, many of which are associated with stress responses. For example, AtMYB2 interacts with calmodulin to enhance salt tolerance in *A. thaliana* [12]. The expression of genes involved in abscisic acid (ABA) synthesis in *A. thaliana* is promoted by MYB15 to enhance salt tolerance and drought tolerance [13]. Studies have shown that an increase in the OsMYB2 expression level can simultaneously increase the salt tolerance of rice [14]. The AtMYB74 gene in *A. thaliana* R2R3-MYB family is transcriptionally regulated by RNA-directed DNA methylation (RdDM) under salt stress to increase salt tolerance [15]. Studies have shown that SlMYB102 participates in multiple signal transduction pathways in response to salt stress, so as to improve the survival rate of plants under salt stress [16]. Salt stress can activate the defense response of plants to stress, among which *AtMYB49* can regulate the keratin deposition in leaves, increase the content of Ca^+^ in leaves, improve the oxidative and antioxidant capacity, and finally improve the tolerance to salt resistance of plants [17].

Abiotic stresses to which plants are often exposed, in addition to salt stress, include low-temperature stress. Under low temperature conditions, the MYB gene can respond to physical and chemical reactions [18]. *A. thaliana* AtMYB15 is a negative regulator of freezing tolerance and inhibits the expression of CBF1/DREB1 to improve cold tolerance [19]. In rice, OsMYBS3 can inhibit the DREB1/CBF-dependent cold signaling pathway, thus improving the cold tolerance of rice [20]. The rice OsMYB3R-2 gene is induced by cold stress, and its overexpression regulates the cell cycle and improves the cold tolerance of rice plants [21].

At present, on the basis of our previous transcriptome data, we first report that an unknown function MYB transcription factor gene, *FvMYB82*, from diploid strawberry is involved in the salt and cold stress response. Our results indicated that *FvMYB82* plays great positive role in transgenic *A. thaliana* as an R2R3 family member with salt stress and cold stress, and we are the first to find that the *FvMYB82* protein localized onto Nucleus and *FvMYB82* gene respond to CBF and ABA pathway. 

## 2. Results

### 2.1. Cloning and Bioinformatic Analysis of FvMYB82

The full-length *FvMYB82* gene cloned from *Fragaria vesca* was 960 bp and encoded 319 amino acids (Appendix A). The ExPASy ProtParam tool (SIB Swiss Institute of Bioinformatics, Swiss. https://web.expasy.org/protparam/, accessed on 1 June 2022) predicted that the encoded protein had a theoretical molecular weight of 36.116 kDa and a theoretical isoelectric point of 6.64. The Ser (10.7%), Thr (9.1%), Asn (7.5%), and Lys (7.2%) residues predominated in the amino acid composition of the *FvMYB82* protein. The total number of negatively charged residues (Asp + Glu) was 41, whereas the total number of positively charged residues (Arg + Lys) was 39. The overall average hydropathicity of *FvMYB82* was −0.971, indicating that the protein was hydrophilic. 

The *FvMYB82* gene included two MYB conserved DNA-binding domains, consistent with 13 species (Figure 1A). A phylogenetic tree revealed that *FvMYB82* was clustered in the same lineage with *RcMYB82* of *Rosa chinensis*, and thus the two genes showed the highest homology (Figure 1B). Analysis of the secondary structure revealed that the *FvMYB82* protein comprised 24.45% alpha helices, 6.90% beta turns, 12.54% extended strands, and 56.11% random coils (Figure 2A). The *FvMYB82* amino acid sequence contained two SANT-conserved domains (Figure 2B), one located at 16–69 aa and the other located at 72–120 aa. The presence of these domains showed that *FvMYB82* was a member of the R2R3-MYB family. Analysis with the SWISS-MODEL online tool predicted a tertiary structure with a HTH region, consistent with the predicted secondary structure for *FvMYB82* (Figure 2C).

### 2.2. FvMYB82 Was Localized onto the Nucleus

The *FvMYB82* gene was connected to the pCAMBIA2300 vector. The resulting 35S::*FvMYB82*-GFP construct was bombarded onto onion epidermal cells, so as to visualize the subcellular localization of the *FvMYB82* protein. As observed with a confocal laser scanning microscope, the fluorescence of the control 35S::GFP construct was distributed throughout the cell in the nucleus, cell membrane, and cytoplasm, whereas the fluorescence signal of the 35S::*FvMYB82*-GFP fusion protein was localized onto the nucleus. This was consistent with the effect after DAPI staining (Figure 3).

### 2.3. Expression Level Analysis of FvMYB82 in F. vesca Seedlings

The expression patterns of *FvMYB82* in immature leaves, mature leaves, stems, and roots of *F. vesca* were analyzed with quantitative RT-PCR technology. The expression level of *FvMYB82* was highest in immature leaves, but it was much lower in stems and mature leaves (Figure 4A). In immature leaves, low temperature was the first abiotic stress treatment to induce *FvMYB82* expression. The expression level was significantly higher under cold stress than that of the control (CK) and under salt, drought, heat, or ABA treatment. The expression of cold stress reached the peak at 2 h and decreased gradually within 12 h. In the other stress treatments, the expression level in immature leaves in the salt and drought treatments peaked at 8 h. That of the heat treatment peaked at 2 h, and it was highest at 6 h under ABA treatment (Figure 4B). In the roots, cold treatment induced *FvMYB82* expression first, and the expression level was significantly higher than that in the salt, drought, heat, and ABA treatments and the CK. The highest expression level in the roots was detected at 4 h and thereafter decreased within 12 h. The expression level in the roots under salt or ABA treatment peaked at 2 h, that in the drought treatment peaked at 4 h, and that in the hot treatment expression peaked at 6 h (Figure 4C). qPCR analysis of strawberry showed that the expression of *FvMYB82* was relatively significant under salt stress, low-temperature stress, ABA stress and drought stress. We preliminarily conducted salt stress and low-temperature stress.

### 2.4. Overexpression of FvMYB82 in A. thaliana Enhanced Salt Tolerance

Among 12 the transgenic *A. thaliana* plants, only five lines (L1, L2, L3, L4, and L5) were identified as transgenic plants by using the real-time fluorescent quantitative PCR (qPCR), with the wild-type (WT) and unloaded line (UL) lines as controls (Figure 5A). Therefore, T_3_ transgenic *A. thaliana* lines were used as test materials for further analysis. According to the statistics of the germination rates of WT, UL, and transgenic lines, it was found that the germination rates of the five groups of *A. thaliana* plants were not different under normal growth conditions. Under salt stress, the germination rates of WT and UL were 25% and 28%, respectively. The germination rates of L1, L3 and L5 lines were 70%, 68%, and 69%, respectively (Appendix A). 

The response of *FvMYB82* to salt stress was further explored by treating T_3_ transgenic *FvMYB82*-OE *A. thaliana* (L1, L3, and L5), WT, and UL with 200 mM NaCl for 7 days, followed by irrigation with water for 3 days for recovery. Under the non-stress condition (0 d), the transgenic lines (L1, L3, and L5), WT, and UL exhibited an identical phenotype. After 200 mM NaCl treatment for 7 days, the leaves of all lines were yellow, and after recovery (water treatment) for 3 days, the phenotypic recovery of the WT and UL was inferior to that of the transgenic plants (Figure 5B). The percentage survival of the L1, L3, and L5 lines was 78%, 73%, and 75%, respectively, whereas the WT and UL had survival rates of 21% and 32%, respectively. The *FvMYB82*-OE *A. thaliana* showed significantly increased survival under salt stress (Figure 5C). 

All *A. thaliana* lines grown in the control environment and under salt stress were analyzed for physiological indicators. Overexpression of *FvMYB82* in transgenic *A. thaliana* reduced the increase in malondialdehyde (MDA) content in response to salt treatment, whereas the proline content was increased significantly and activities of catalase (CAT), peroxidase (POD), and superoxide dismutase (SOD) were higher in the transgenic overexpression lines than the WT under salt stress (Figure 6). The chlorophyll content was decreased in all lines under salt stress and was lower in the WT and UL compared with the L1, L3, and L5 lines. The EL of WT and UL was significantly higher than that of transgenic type under salt stress (Appendix A). These results indicated that the high expression of *FvMYB82* in *A. thaliana* could enhance plants’ resistance to high salt stress.

### 2.5. Overexpression of FvMYB82 Improved the Expression Levels of Salt Tolerance-Related Genes

Salt tolerance is dependent on the ABA signaling pathway. Therefore, expression analysis of *AtSnRK2.4* and *AtSnRK2.6*, two genes involved in ABA signal transduction [22,23], in *A. thaliana* overexpressing *FvMYB82* was performed (Figure 7). The expression levels of *AtSnRK2.4* and *AtSnRK2.6* in each *A. thaliana* line did not differ in the non-stress environment. However, under salt stress, the expression levels of *AtSnRK2.4* and *AtSnRK2.6* in WT plants were much lower than those in *FvMYB82*-overexpressing plants, indicating that *FvMYB82* may function as an upstream transcription factor that positively regulated the downstream genes *AtSnRK2.4* and *AtSnRK2.6* under salt stress. These results also indicated that ABA signal transduction was responsive to salt stress. Kup6 potassium transporter plays a role in ABA biosynthesis during osmoregulation under the control of ABA [24]. ABA biosynthesis in *A. thaliana* requires an ABA distal element of the *AtNCED3* promoter [25]. In the meantime, the expression levels of *AtKUP6* and *A**tNCED3* were significantly higher under salt stress than under the non-stress condition, and the expression levels were significantly higher in the transgenic *A. thaliana* lines than the WT. These findings suggested that *FvMYB82* modulated crucial genes involved in plant salt stress and enhanced plant salt tolerance.

### 2.6. Overexpression of FvMYB82 Improved the Cold Tolerance of A. thaliana

The 4-week-old seedlings of the WT, UL, and T_3_ *FvMYB82*-OE *A. thaliana* (L1, L3 and L5) were cultured at 4 °C for 48 h in an incubator under a 14 h/10 h (light/dark) photoperiod, then subjected to −6 °C freezing treatment for 14 h, and then returned to the normal environment (14 h, 22 °C/10 h 18 °C (light/dark)) for 7 days. The phenotype of the *FvMYB82*-overexpressing *A. thaliana* plants was almost identical to that of the WT and UL in the control environment. However, cold treatment caused an obvious difference in phenotype among these lines (Figure 8A). Compared with transgenic L1, L3 and L5, the leaves of WT and UL plants were more damaged and difficult to recover. The percentage survival of the three *FvMYB82*-overexpression lines was 78%, 79%, and 77%, whereas that of the WT was 25% and the UL was 30% (Figure 8B). Meanwhile, with cold treatment, the germination rates of WT and UL were 30% and 26%, respectively. The germination rates of L1, L3, and L5 lines were 73%, 74%, and 68%, respectively (Appendix A).

Physiological indicators associated with cold stress in transgenic *A. thaliana* lines (L1, L3, and L5) were analyzed. Almost no differences in MDA, chlorophyll, and proline contents, and POD, SOD, and CAT activities among these lines and the WT under the control condition (22 °C) (Figure 9). The activities of POD, SOD, and CAT and the proline content increased after cold treatment, and the proline content increased to a greater extent in transgenic *A. thaliana* compared with the WT. The content of MDA increased under cold treatment, but the degree of increase was alleviated in the transgenic lines compared with the WT. The chlorophyll content of *A. thaliana* overexpressing *FvMYB82* and the WT decreased under cold treatment, but the decline was strongly alleviated in the transgenic lines. The EL of WT and UL was significantly higher than that of transgenic type under cold stress (Appendix A). In conclusion, a high level of *FvMYB82* expression in *A. thaliana* may increase its resistance to low temperatures. 

### 2.7. Overexpression of FvMYB82 Enhanced the Expression Levels of Cold Tolerance-Related Genes

The CBF (CRT-Binding Factor/DRE-Binding Protein) transcription factor family is commonly used as an indicator of cold response pathways in *A. thaliana* [26]. Therefore, the expression of several cold-responsive transcription factors downstream of *MYB* genes (*At**CBF1*, *AtCBF2*, *AtCOR15a*, and *AtCOR78*) under cold treatment was analyzed in *FvMYB82*-overexpression *A. thaliana* lines and the WT (Figure 10). The plants were treated at −6 °C for 14 h. The expression levels of *AtCB**F1*, *AtCBF2*, *AtCOR15a*, and *AtCOR78* in the WT and UL were significantly lower than those in the *FvMYB82*-overexpression lines, indicating that *FvMYB82*, as an upstream transcription factor, positively regulated the downstream genes *At**CBF1* and *AtCBF2*. Subsequently, the expression levels of *AtCOR15a* and *AtCOR78* were increased, and ultimately the cold tolerance of the plants was greatly improved.

## 3. Discussion

The composition of MYB transcription factors in plants contains a conserved domain, which is composed of one to four incomplete repeated amino acid sequences (R1, R2, and R3), which is called the R structure. The largest family of MYB proteins in plants is the R2R3-MYB family, which has two conserved domains [27]. In the present study, wild-type forest strawberry was used as the experimental material and the sequence of *FvMYB82* was identified from the strawberry gene bank (http://bioinformatics.towson.edu/strawberry/newpage/TF_Clustering.aspx (accessed on 22 September 2020)). Primers were designed using DNAMAN and the gene sequence was cloned. The length of the *FvMYB82* nucleotide sequence was 960 bp, encoding 320 amino acids, and the predicted protein had an average hydrophilicity of 1.02, and thus was predicted to be a relatively hydrophilic protein. *FvMYB82* contained two SANT-MYB DNA-binding domains, which are conserved domains typical of R2R3-MYB family members; thus, it was speculated to be a typical R2R3-MYB gene. Multiple sequence alignment between *FvMYB82* protein sequence and MYB protein sequence of other species showed that *FvMYB82* protein and other MYB family proteins had high similarity in the conserved sequence, but there was a big difference in the non-conserved region, which was consistent with the characteristics of transcription factors. The two SANT-conserved domains showed similarity to MbMYB4 from *Malus baccata* [28]. A phylogenetic tree revealed that *FvMYB82* was phylogenetically closest to RcMYB82.

Previous studies have reported that transcription factors mainly function in the nucleus of cells. For example, the MbMYB4 protein is localized to the nucleus [28]. Using the particle bombardment method, the *FvMYB82*-GFP fusion vector was inserted into onion epidermal cells for transient expression and observed with a confocal microscope. As a control, the green fluorescent protein (GFP) was distributed throughout the whole cell, while the fluorescence of the *FvMYB82*-GFP fusion protein was only localized onto the nucleus (Figure 3E). DAPI staining of the cell nucleus can further identify that *FvMYB82* protein was located onto the cell nucleus. These results were consistent with those for *M**. baccata* [28]. However, the subcellular localization of certain MYB proteins may differ in other plant species. The tomato SlMYBl protein was localized in the mitochondria [29], and in *Prunus campanulata,* PcMYB was localized on the plasma membrane [30].

The MYB transcription factors play important roles in plant growth processes. Some MYB genes regulate growth and development, and response to pathogen attack; some are involved in tolerance to cold, salt, drought, and other stresses; and some participate in plant morphogenesis. In *A. thaliana*, *AtMYB58*, *AtMYB63*, and *AtMYB85* activate lignin synthesis in fibers and vascular bundles, thus improving the ability to resist stress [31]. *A. thaliana AtMYB26* controls secondary wall deposition in the anther [32]. Trichomes are important epidermal structures in plants. Trichomes can reduce external environmental stress during plant growth and development and play a crucial role in plant stress responses [33]. The *AtMYB82* transcription factor positively regulates trichome formation in *A. thaliana*, and the increase in trichome density correspondingly increases tolerance of salt and cold stress [34].

The expression pattern of *FvMYB82* was indicated to be tissue specific in *F. vesca*. The expression level of *FvMYB82* was highest in immature leaves and lowest in mature leaves of *F. vesca*, and the expression level in immature leaves was nine times that of mature leaves. The results showed, in accordance with *MbMYB4 and MbMYB108* from *M.baccata*, that the *MYB* gene expression level in immature leaves were higher than in other tissues, indicating the new leaves may be more sensitive to external stress. For these reasons, young leaf and root were chosen for further identification. The present results showed that low temperature, salinity, drought, high temperature, and ABA induced high-expression levels of *FvMYB82* in *F. vesca*, and that the expression levels varied with treatment duration. In a cold environment, the expression level of *FvMYB82* was distinctly elevated. It peaked at 2 h in young leaf and peaked at 4h in root. Under salt stress, *FvMYB82* expression peaked at 8 h in young leaf and peaked at 2 h in root, and in the young leaf, the maximum level was 4.6 times that of the control. The expression level of *FvMYB82* under drought stress peaked at 8 h in young leaf and peaked at 4 h in root. The maximum was 9.2 times that of the control in young leaf and 9.1 times in root. These results indicated that *FvMYB82* in different parts of the plant was responsive to cold, salt, and ABA treatment, which provided the basis for the subsequent analyses.

When plants are under stress, their cell plasma membranes become more permeable, electrolytes and soluble substances leak out, then EL increases. Therefore, the measurement of relative EL is a method used to determine the resistance of plants to stress [28]. In this study, under salt stress and cold stress, the germination rates of *A. thaliana* overexpressing *FvMYB82* (L1, L3 and L5) were significantly higher than WT and UL, and the relative EL of (L1, L3 and L5) was significantly higher than that of the wild type. Thus, overexpression of *FvMYB82* increased the tolerance of *A. thaliana* to salt and cold stresses. The contents of proline, chlorophyll, and MDA and activities of antioxidant enzymes are indicators of stress-induced damage to plants [35,36,37]. Proline can change the osmotic concentration of cells, which acts as an important osmotic regulator of plant cells. In this study, the contents of proline increased, becoming significantly higher than WT and UL, which showed that the salt-stress-induced transgenic lines improved the salt tolerance obviously. The increase in MDA content is an indicator of increased oxidative damage to membranes in plant cells [38], when WT, UL, and three lines (L1, L3, and L5) were under salt stress, MDA content of WT and UL was significantly higher than transgenic lines, indicating that WT and UL membranes were damaged more seriously. Under high salt stress, damage to the leaf cells results in reduction in the chlorophyll content, and the leaf color changes to yellow. Therefore, analysis of the chlorophyll content in the leaves can reveal the stress resilience of plants [39]. In this study, under salt stress, the leaves of WT and UL showed more yellow than *FvMYB82*-OE *A. thaliana*. Antioxidant enzymes eliminate excess oxidation products and protect cells from oxidative damage [40]. For these, the activities of antioxidant enzymes in plants indicates the degree of plant tolerance to stress. Under salt stress, the contents of CAT, SOD, and POD in transgenic lines (L1, L3, and L5) increased, becoming higher than WT and UL, indicating that *FvMYB82*-OE improved salt tolerance.

The ABA signaling pathway is responsive to salt stress. Previous studies have shown that the ABA content increases in plants under salt stress, and improved tolerance to salt stress is associated with an increase in ABA content. The ABA response elements (ABRE) and ABRE-binding proteins/ABRE-binding factors (AREB/ABFs) are critical in ABA-dependent gene expression during stress response [41]. *A. thaliana*
*AtSnRK2.6* and other genes respond to ABA-dependent and ABA-independent pathways, whereas *AtSnRK2.10* responds only to ABA-independent pathways [42]. The SnRK2 family plays a crucial role in the ABA signal transduction pathway. SnRK2 kinase activity is inhibited in the absence of ABA. Under salt stress, ABA receptor proteins, including PYR1, PyR1-like protein (PYL), and ABA receptor regulatory component (RCAR), simultaneously accumulate ABA in response to salt stress, thereby inhibiting the phosphatase activity of PP2C. In addition, the kinase activity of SnRK2.4 activated by salt stress is inhibited by ABI1 and acid-sensitive phosphatases of the phosphoprotein phosphatase (PPP) family. It has been concluded that, under the influence of salt stress, ABA is accumulated, which activates SnRK2.4 and inhibits type 2C PP2CA protein phosphatase and ABI1, thus regulating the plants’ response to salt stress [43]. In addition, ABA can regulate plant tolerance to stress by regulating stomatal closure. Under exposure to stress, the increase in ABA content activates *SnRK2.6*/*OST1*, a key regulator of Ca^2+^-independent stomatal closure, and OST1 activates *SLAC1*, *KUP6*, and *QUAC1*. The exosmosis of Cl^−^, K^+^, and Malate^2−^ is further promoted, whereas internal K^+^ transport is reduced. Ultimately, ABA regulates plant adaptation to stress by inducing stomatal closure through ion flux in the guard cells [44]. Under drought stress, the ABA content in plants will increase, which will induce stomatal closure to regulate water loss. In *A. thaliana*, *AtNCED3* is a crucial gene for ABA synthesis [45]. *AtNCED3* is mainly expressed in the vascular bundles of leaves [46]. The synthesis and accumulation of ABA in the leaves has a regulatory effect on stomatal closure, and thus increases the salt tolerance of plants.

Transformation with *MbMYB4* improves the tolerance of transgenic *A. thaliana* plants to cold stress [47]. In the present study, WT, UL, and *FvMYB82*-transgenic *A. thaliana* lines (L1, L3, and L5) suffered damage under low temperature and high salt stress, but the transgenic lines showed less severe yellowing, and growth was restored after they were transferred to a non-stress environment. The percentage survival of the transgenic lines was significantly higher than that of the WT and UL plants. These results showed that *FvMYB82* significantly improved the resilience of *A. thaliana* to low temperature and high salt stress. To examine the mechanism by which *FvMYB82* improved the stress tolerance of *A. thaliana*, physiological indicators and downstream gene expression of stress-treated *A. thaliana* lines were analyzed.

Proline is an important osmotic regulator of plant cells, which can change the osmotic concentration of cells, reduce the freezing point, and increase the cold tolerance of plants [48]. Under low-temperature stress, osmotic regulators such as proline will accumulate rapidly in plants [49]. The accumulation of proline increases the water retention of cells, promotes hydrin formation, and increases the content of soluble protein in cells, so that plants can better resist the effects of low temperature. The proline content of the transgenic *A. thaliana* lines (L1, L3, and L5) was increased under cold stress, indicating that the transgenic lines had better water retention than the WT and UL, and a superior ability to withstand cold stress. Chlorophyll is an important indicator of the cold tolerance of plants. A low temperature will result in chlorophyll degradation in the leaves, causing yellowing and disruption of photosynthesis [50]. Under low-temperature stress, the leaves of the transgenic lines (L1, L3, and L5) showed less severe yellowing, whereas the leaves of the WT and UL experienced more severe yellowing. Thus, the chlorophyll content in the transgenic lines was higher than that in the WT and UL under cold stress. These results indicated that the transgenic lines could more effectively resist chlorophyll decomposition at a low temperature. Oxidative products produced by plants in response to low temperature and high salt stress are removed by the reactive oxygen species scavenging system, which includes SOD, CAT, and POD, so as to avoid oxidative damage and improve stress tolerance [51]. The POD and CAT activities in rice roots under cold stress are significantly increased compared with those in non-stressed rice [52]. In the present study, compared with the WT and UL, *A. thaliana* overexpressing *FvMYB82* had higher activities of CAT, POD, and SOD, and lower contents of MDA under cold stress. These results indicate that transgenic *A. thaliana* has better antioxidant ability and less damage to plants at low temperature.

MYB gene can bind to CBF promoter, thereby promoting the expression of various downstream genes associated with cryogenic stress, inhibiting CBFs response, and further negatively regulating the tolerance of plants to low-temperature stress [53,54,55]. In addition, the crucial genes responsive to the CRT/DRE pathway, such as CORs, RDs, LTIs, and other cis-acting elements, can be induced by CBFs, thus positively enhancing the ability of plants to cope with cold stress. MYB transcription factors can also negatively regulate freezing tolerance. For example, *MYB15* functions as a negative signaling regulator and is degraded by *PUB25* and *PUB26*, the targets of the protein kinase *OST1*. Low temperature activates OST1 to phosphorylate *PUB25* and *PUB26*, which enhances their activity, and promotes the ubiquitination and degradation of MYB15, thus positively inducing expression of ICE1 and CBFs at low temperature [56,57,58,59]. In this study, *FvMYB82* positively regulates cold tolerance through two key genes *At**CBF1* and *AtCBF2*, as well as the expression of downstream cold responsive genes *AtCOR15a* and *AtCOR78*.

Based on the above results and former studies, we made a potential model to describe the role of *FvMYB82* in salt and cold stresses (Figure 11). The expression levels of SnRK family genes (*SnRK2.4* and *SnRK2.6*), *AtNCED3* and *AtKUP6* increased obviously in transgenic (L1, L3 and L5) *A. thaliana* compared with WT and UL lines under salt stress, indicating that *FvMYB82* could promote downstream salt-stress-related gene expression and positively regulate the ABA biosynthesis and signal transduction pathway to improve the slat tolerance of plants. ABA and drought stress increased *SsMYB113* transcription, further improving expression of the corresponding biosynthetic genes *SsNCED* in overexpressed *SsMYB113 A. thaliana* lines. At the same time, ROS of *SsMYB113*-OE were modulated to regulate drought stress tolerance [60]. For this, we proposed that the enzyme of transgenic plant was modulated by ABA signal to improve the abiotic stress tolerance in the hypothesis model. Meanwhile, in the cold environment, the CBF family genes (*CBF1* and *CBF2*), *COR15a* and *COR78* expression levels were higher in transgenic (L1, L3, and L5) *A. thaliana* than in WT and UL, indicating that the cold stress could induce the expression of *FvMYB82*, and *FvMYB82* domain binding to the promoter regions of *CBF1* and *CBF2* and promote binding of CBFs to the CRT/DRE cis-acting elements of downstream genes, thereby activating expression of the downstream cold-responsive genes *COR15a* and *COR78*, finally enhancing plant cold tolerance.

## 4. Materials and Methods

### 4.1. Plant Materials, Growth Conditions, and Treatment

The diploid *F. vesca* seedlings were cultured on Murashige and Skoog (MS) medium supplemented with 0.6 mg/L indole-3-butyric acid (IBA) and 0.6 mg/L 6-benzylaminopurine (6BA) or in soil in an incubator under a 16 h/8 h (light/dark) photoperiod and 70% relative humidity at 22 °C at the College of Horticulture and Landscape Architecture, Harbin, China. Seeds of *A. thaliana* ecotype Columbia-0 were obtained from the *A. thaliana* Biological Resource Center (https://abrc.osu.edu/, Ohio State University, USA (accessed on 16 October 2019)). *A. thaliana* seedlings cultured in an incubator under the same conditions as wild strawberry. For low-temperature treatment, strawberry seedlings were grown in an incubator at 4 °C. To test the drought, salt, and ABA stress tolerances, the seedlings were treated with 15% PEG6000, 200 mM NaCl, or 100 μM ABA, respectively [61]. *F**. vesca* materials treated with abiotic stresses were sampled at different time intervals (0, 1, 2, 4, 6, 8, and 12 h). The collected immature leaves, mature leaves, stems, and roots were quickly frozen with liquid nitrogen and stored at −80 °C for tissue-specific gene expression analysis [62].

### 4.2. Isolation and Cloning of FvMYB82

Total RNA was extracted from the *F. vesca* materials (immature and mature leaves, stems, and roots) using the OminiPlant RNA Kit (Conway Collection, Beijing, China) following the manufacturer’s instructions. The TransScript First-Strand cDNA Synthesis SuperMix (TransGen Biotech, Beijing, China) was used to synthesize the first-strand cDNA. Agarose gel (1%) electrophoresis was used to analyze and verify the RNA and cDNA. The coding sequence of *F. vesca Fv**MYB82-like* (XM_004288737.2) was used as the reference sequence to amplify the full-length cDNA sequence, and gene-specific primers (*FvMYB82*-F and *FvMYB82*-R; Appendix A) were designed with Primer 5.0 software. The PCR product fragments were purified and ligated into the ASY-T1 vector (TransGen Biotech, Beijing, China) for sequencing [35].

### 4.3. Subcellular Localization of FvMYB82

The coding region of *FvMYB82* was amplified and inserted into the pSAT6-GFP-N1 vector to generate fusion proteins with the green fluorescence protein (GFP). The upstream and downstream primers (*FvMYB82*-slF and *FvMYB82*-slR; Appendix A) together with *Sal*I and *BamH*I restriction enzymes were used to obtain *FvMYB82* gene fragments. *Sal*I and *BamH*I acted as double-digestion restriction enzymes to digest the PCR product and the pSAT6-GFP-N1 vector. The *FvMYB82* fusion plasmid containing the target fragment was then inserted into onion ‘Yachunya57′ outer epidermal cells using the particle bombardment method. The empty 35S::GFP vector was used as a control [63]. The subcellular expression of the *FvMYB82*-GFP fusion protein was observed with a confocal microscope (Olympus Corporation, Tokyo, Japan). DAPI staining was used as a nucleus marker for nucleus detection.

### 4.4. Sequence Analysis and Structure Prediction of FvMYB82

In order to achieve multiple sequence alignment of *FvMYB82* and other species of MYB TFs, DNAMAN 5.2 was performed. The phylogenetic tree was constructed using a neighbor-joining method [64] with MEGA7 (available online: http://www.megasoftware.net (accessed on 1 June 2022)) [65]. To predict the primary structure of *FvMYB82* protein, ExPASy was used as a tool (available online: https://web.expasy.org/protparam/(accessed on 1 June 2022)). Using SMART (SMART: Main page. Available online: http://smart.embl-heidelberg.de/ accessed on 1 June 2022) website to predict the domain of *FvMYB82* protein, the tertiary structure of *FvMYB82* protein was predicted on the SWISS-MODEL website (SWISS-MODEL. Available online: https://swissmodel.expasy.org/ accessed on 1 June 2022) [66].

### 4.5. Expression Analysis of FvMYB82

The qPCR (Appendix A) was used to detect the expression levels of *FvMYB82* under multiple abiotic stresses in different tissues using the primers *FvMYB82*-qF and *FvMYB82*-qR (Appendix A). The PCR protocol was as follows: 30 s at 94 °C; then 40 cycles of 5 s at 95 °C, 40 s at 54 °C, and 30 s at 72 °C; and 10 min at 72 °C. The expression of the *FvMYB82* gene was detected with the TB Green™ Premix Ex Taq™ II (Tli RNaseH Plus kit) (TaKaRa, Beijing, China) in accordance with the manufacturer’s protocol [67]. The *Actin* gene (XM_011471474.1, *F. vesca*) was used as the internal reference gene. The *FvMYB82* expression level was calculated using the 2^−∆∆*C*t^ method.

### 4.6. Stress Treatment and Determination of Related Physiological Indexes in A. thaliana

For the germination rate, *A. Thaliana* wild-type (WT), no-load system (UL) and T3 transgenic lines (L1, L3, L5) were divided into two groups, respectively. For cold treatment, one group was planted in 1/2 MS medium, after being vernalized in a 4 °C refrigerator for two days. The Petri dish was cultured in a −6 °C refrigerator. The other group was planted in a Petri dish with filter paper for salt treatment, and 10 mL 200 mM NaCl solution was added to one dish. After being vernalized at 4 °C, these Petri dishes were transferred into a light incubator, with 50 seeds per line, three replicates every treatment. Germination was observed after one week.

For related Physiological Indexes in *A. thaliana*, wildtype (WT), empty vector line (UL) and T3 transgenic lines (L1, L3, L5) of *A. thaliana* were planted in1/2 MS medium for 10 days. When cotyledons showed, the seedlings were transferred to nutrient pots containing a substance including soil and vermiculite (2:1) (4 plants per pot). Every line of *A. thaliana* was divided into two groups: one group was treated with salt stress (irrigating 200 mM NaCl for 7 days), and then water for 3 days for recovery. The other group was treated with cold stress (14 h, −6 °C), and later cultured in normal condition for 7 days for recovery. Morphological characteristics were observed, and the percentage survival was determined [68].

All materials of each line treated with different stress and CK were collected for a physiological indexes test. The fresh leaf samples were soaked in the mixture of ethanol and acetone for 24 h, and then the absorbance of the mixture at 645 nm and 663 nm was measured using a spectrophotometer, which was used to calculate the content of chlorophyll [69]. The content of MDA was determined according to the reaction of MDA with thiobarbituric acid under acidic and high temperature conditions to generate red–brown trimethyloxazole, and the maximum absorption peak was at 532 nm. [70]. Refer to the method of Huang et al. the content of proline was determined by sulfosalicylic acid method [71,72,73]. The activity of SOD was determined according to its inhibition of NBT reduction under light. [74], In the presence of hydrogen peroxide, POD can oxidize guaiacol to produce a brown color, and the product can be determined by a spectrophotometer. [75] Catalase can decompose hydrogen peroxide so that the absorbance (A240) of reaction solution decreases with the reaction time. Catalase activity can be measured according to the change rate of absorbance. [76]. We measured the conductance of the extract solution with a conductivity meter, heated it in a boiling water bath for 30 min, cooled it down sufficiently, measured the conductance again, and used the formula to calculate the EL [77].

### 4.7. Expression Analysis of Genes Associated with Salt Tolerance in FvMYB82-OE A. thaliana

Using *AtActin* as an internal reference, the total mRNAs were extracted from untreated (CK) and salt-treated *A. thaliana* lines (WT, UL, L1, L3, and L5). The cDNA was used as a template for reverse transcription of the first-strand cDNA. The expression levels of four salt stress-responsive genes (*AtSnRK2.4*, *AtSnRK2.6*, *AtKUP6*, and *AtNCED3*) located downstream of MYB transcription factors were quantified by qPCR analysis [78,79]. The specific primers used are listed in Appendix A. The procedure followed the reaction system described in Section 4.5.

### 4.8. Expression Analysis of Cold-Tolerance-Related Genes in A. thaliana Overexpressing FvMYB82

Using *AtActin* as an internal reference, the total mRNAs were extracted from untreated (CK) and low-temperature-treated *A. thaliana* lines (WT, UL, L1, L3, and L5). The cDNA was used as a template for reverse transcription of the first-strand cDNA. Four cold-responsive genes (*At**CBF1*, *AtCBF2*, *AtCOR15a*, and *AtCOR78*) located downstream of MYB transcription factors were quantified by qPCR analysis. The gene-specific primers used are listed in Appendix A. The procedure followed the reaction system described in Section 4.5.

### 4.9. Statistical Analysis

SPSS 21.0 software (IBM, Chicago, IL, USA) was used to analyze the differences via Duncan’s multiple range tests. The bars showed the mean and standard deviation (SD) of three replicates. Statistical differences were referred to as significant when * *p* ≤ 0.05 and ** *p* ≤ 0.01.

## 5. Conclusions

In this study, *FvMYB82* of *F. vesca* was identified and cloned. Overexpression of *FvMYB82* significantly enhanced the tolerance of *A. thaliana* to abiotic stress. The activities of the antioxidant enzyme system were significantly increased, and the membrane–lipid stability and ROS homeostasis were maintained under salt and cold stress. Meanwhile, the morphogenesis of *A. thaliana* seedlings was altered to improve the resistance to stress. The present results show that *FvMYB82* plays a very important role in plants’ response to low temperature and salt stresses.

## Figures and Tables

**Figure 1 ijms-23-10538-f001:**
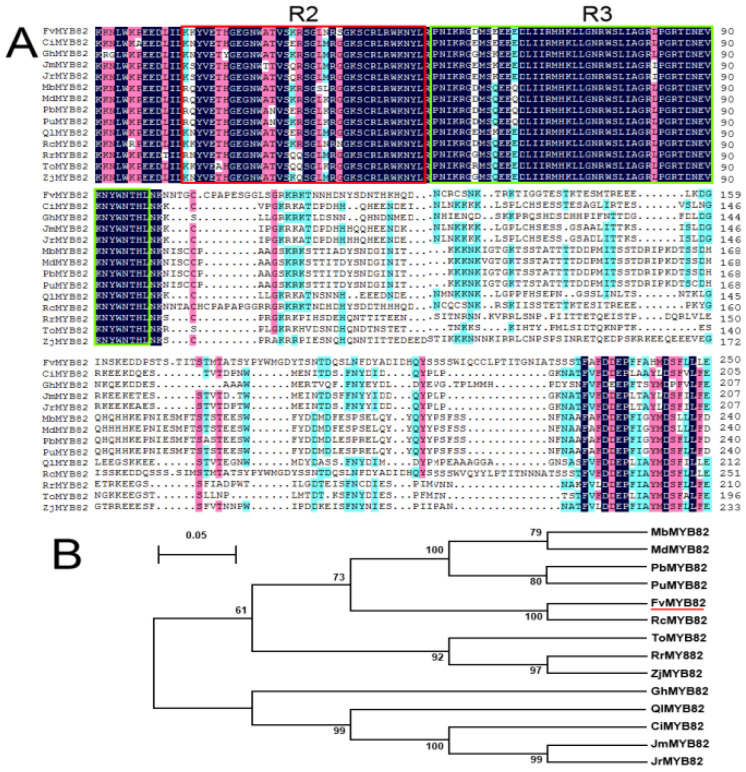
Contrast and evolutionary relationship between *FvMYB82* and MYB transcription factors of different varieties. (**A**) Comparison between homology of *FvMYB82* protein and MYB protein in other plants. The conserved regions of the amino acid sequence are marked by black and red boxes. (**B**) Phylogenetic tree analysis of MYB protein in *FvMYB82* and other plants. The accession numbers are as follows: RcMYB82 (*Rosa chinensis*, XP_024181819.1), MdMYB82 (*Malus domestica*, RXI05710.1), PuMYB82 (*Pyrus ussuriensis*, KAB2620502.1), PbMYB82 (*Pyrus bretschneideri*, XP_009363992.1), MbMYB82 (*Malus baccata*, TQE02283.1), JrMYB82 (*Juglans regia*, XP_018814279.1), ZjMYB82 (*Ziziphus jujuba*, XP_015867230.1), QlMYB82 (*Quercus lobata*, XP_030931205.1), CiMYB82 (*Carya illinoinensis*, KAG6718834.1), RrMYB82 (*Rhamnella rubrinervis*, KAL3453734.1), ToMYB82 (*Trema orientale*, PON99826.1), GhMYB82 (*Gossypium hirsutum*, XP_016720794.1), JmMYB82 (*Juglans microcarpa*, XP_041006667.1).

**Figure 2 ijms-23-10538-f002:**
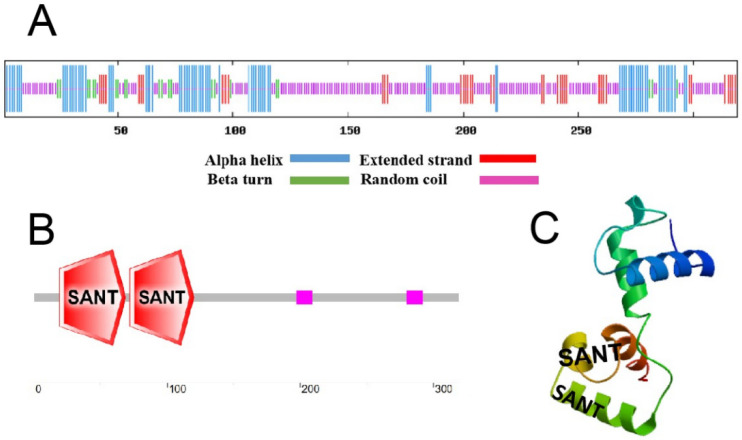
Prediction of *FvMYB82* protein domains and structure. (**A**) Predicted protein secondary structure; (**B**) predicted protein domains; (**C**) predicted tertiary structure.

**Figure 3 ijms-23-10538-f003:**
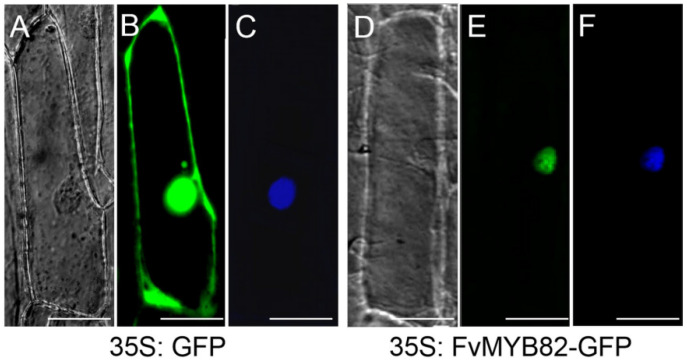
Subcellular localization of *FvMYB82* in onion leaf epidermal cells. The 35S:GFP and 35S:*FvMYB82* plasmids were transformed into the cells by particle bombardment. (**A**,**D**) Bright-field images, (**B**,**E**) GFP fluorescence, (**C**,**F**) cells stained with DAPI. Bar = 50 μm.

**Figure 4 ijms-23-10538-f004:**
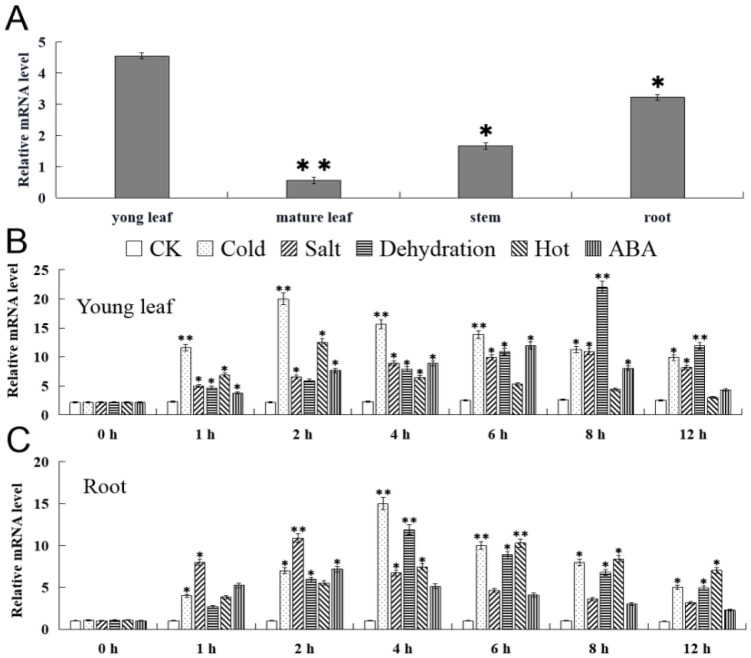
Expression pattern analysis of *FvMYB82* in *F*. *vesca* by quantitative RT-PCR. (**A**) Expression of *FvMYB82* in different tissues in the non-stress environment. (**B**,**C**) Time course of *FvMYB82* expression in young and root in the control and under treatment with salt (200 mM NaCl), heat (30 °C), cold (4 °C), dehydration (15% PEG6000), and abscisic acid (50 μM ABA). Error bars indicate the standard deviation. Asterisks above the error bars indicate a significant difference between the treatment and control (Student’s *t*-test; * *p* ≤ 0.05, ** *p* ≤ 0.01).

**Figure 5 ijms-23-10538-f005:**
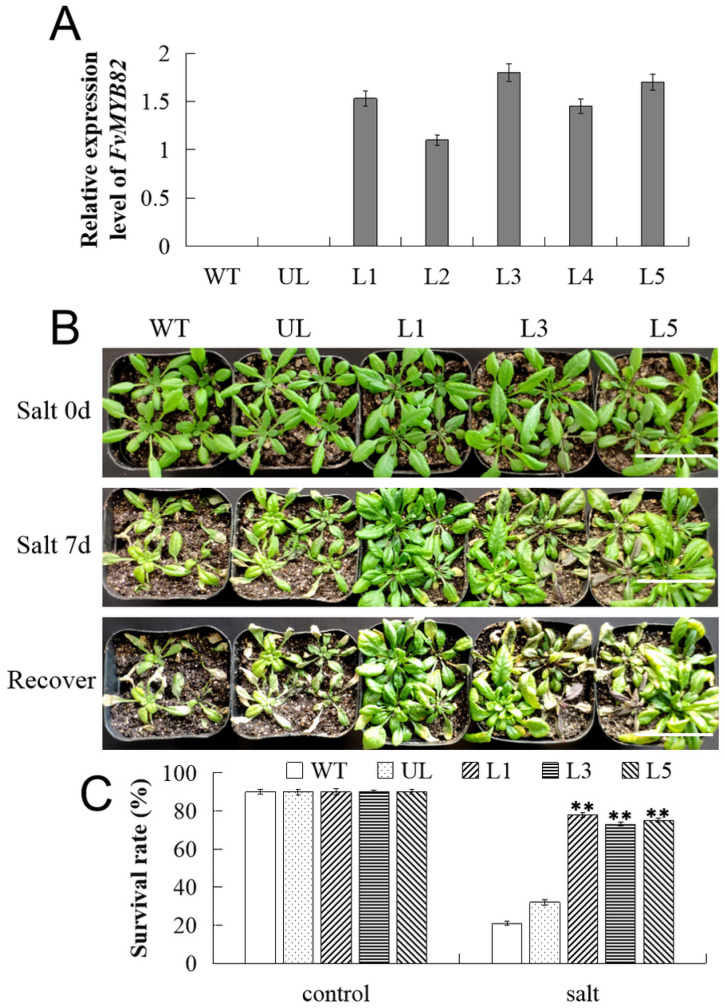
Growth of transgenic *A. thaliana* lines overexpressing *FvMYB82* under salt treatment. (**A**) Relative expression level of *FvMYB82* in WT, UL and 5 *FvMYB82*-overexpression lines (L1, L2, L3, L4 and L5). (**B**) Phenotypes of the WT, UL, and transgenic lines (L1, L3, and L5) grown in the control environment, salt treatment (irrigation with 200 mM NaCl for 7 days), and recovery after salt treatment (irrigation with water for 3 days). Bar = 5 cm. (**C**) Survival percentages of WT, UL, and transformed lines (L1, L3, and L5). Asterisks indicate significant differences between WT and UL, transformed lines (** *p* ≤ 0.01).

**Figure 6 ijms-23-10538-f006:**
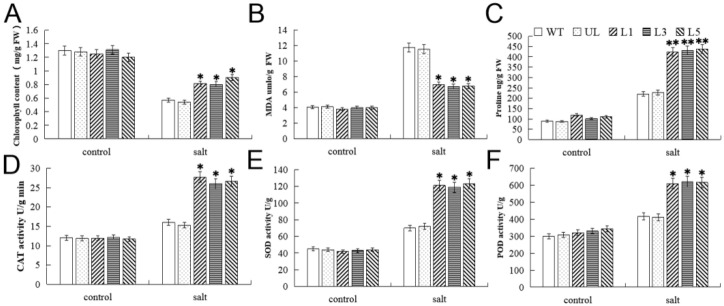
Physiological indicators in transgenic *A. thaliana* lines overexpressing *FvMYB82* under salt treatment. Contents of (**A**) Chlorophyll, (**B**) MDA, (**C**) proline, and the activities of (**D**) CAT, (**E**) SOD, and (**F**) POD in the WT, UL, and *FvMYB82*-OE lines (L1, L3, and L5) under 200 mM NaCl treatment for 7 days. Significant differences were marked with asterisks above the error bar (* *p* ≤ 0.05, ** *p* ≤ 0.01). The levels of indicators in the WT were used as the control.

**Figure 7 ijms-23-10538-f007:**
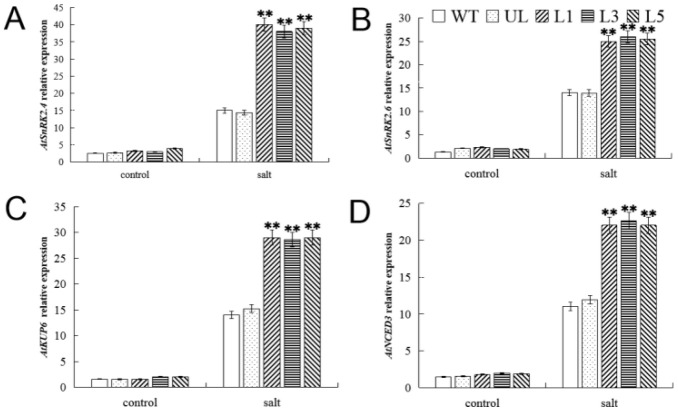
Expression levels of salt-related genes in WT, UL, and transformed *A. thaliana*. overexpressing *FvMYB82* under salt stress. Relative expression levels of (**A**) *AtSnRK2.4*, (**B**) *AtSnRK2.6*, (**C**) *AtKUP6*, and (**D**) *AtNCED3* in the WT, UL, and *FvMYB82*-OE lines (L1, L3, and L5). Data are the average of three replicates. Significant differences are marked with an asterisk above the error bar (** *p* ≤ 0.01).

**Figure 8 ijms-23-10538-f008:**
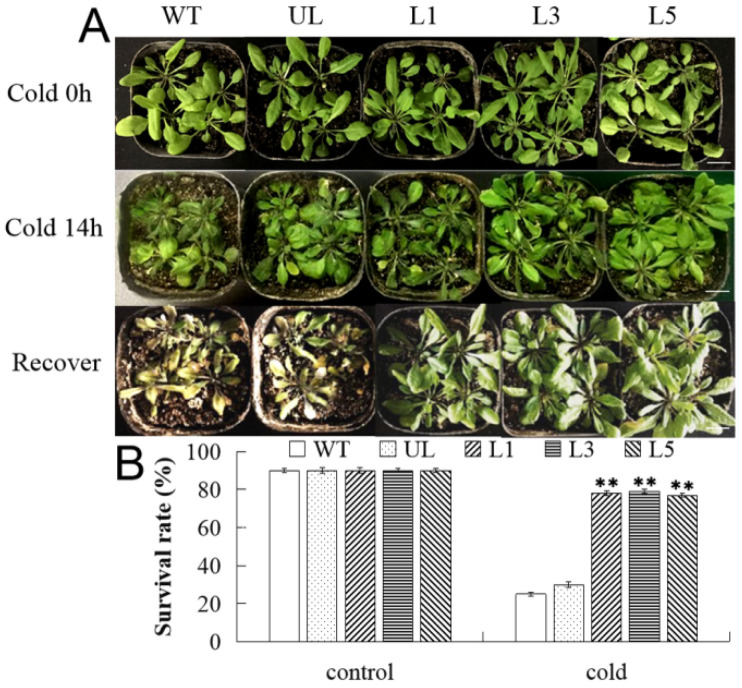
Growth of transgenic *A. thaliana* overexpressing *FvMYB82* under low-temperature treatment. (**A**) Phenotypes of WT, transformants with empty vector (UL), and *FvMYB82*-overexpressing lines (L1, L3, and L5) under the control environment (22 °C), cold treatment (4 °C), and after recovery. Bar = 1 cm. (**B**) Survival rate of WT, UL, and transgenic lines under the control environment and cold treatment. Three replicates were performed. Asterisks indicate a significant difference between the different lines (** *p* ≤ 0.01).

**Figure 9 ijms-23-10538-f009:**
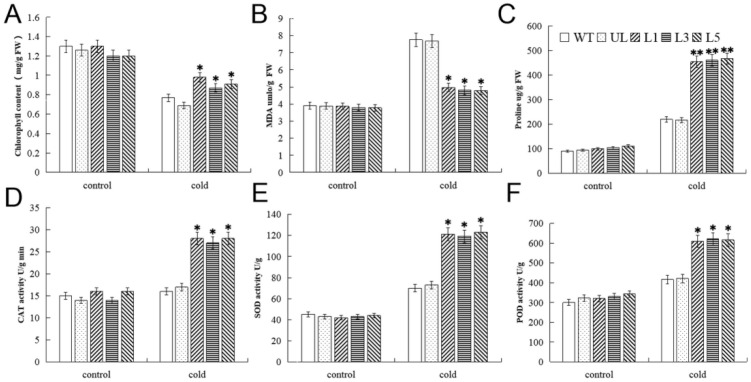
Physiological indicators in transgenic *A. thaliana* lines overexpressing *FvMYB82* under low-temperature treatment. (**A**) Proline content, (**B**) MDA content, (**C**) chlorophyll content, (**D**) CAT activity, (**E**) SOD activity, and (**F**) POD activity in the WT, UL, and *FvMYB82*-overexpressing lines (L1, L3, and L5) under the non-stress environment (22 °C) or cold treatment (4 °C for 12 h). Asterisks above each error bar indicate obviously significant differences between transgenic lines (L1, L3, and L5), UL and the WT (* *p* ≤ 0.05, ** *p* ≤ 0.01). The levels of indicators in the WT were used as the control.

**Figure 10 ijms-23-10538-f010:**
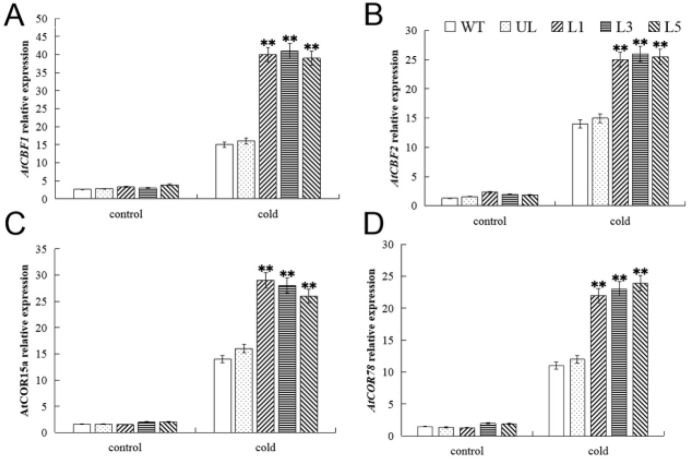
Expression of chilling-related genes in transgenic *A. thaliana* lines overexpressing *FvMYB82* under low-temperature treatment. Relative expression levels of (**A**) *AT**CBF1*, (**B**) *AtCBF2*, (**C**) *AtCOR15a*, and (**D**) *AtCOR78* in the WT, UL, and transgenic lines (L1, L3, and L5). Data are the average of three repetitions. Asterisks indicate extremely significant differences between the transgenic line and the WT (** *p* ≤ 0.01).

**Figure 11 ijms-23-10538-f011:**
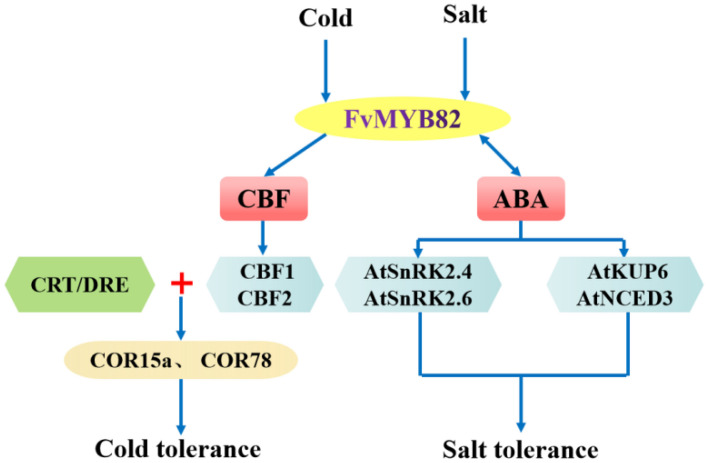
Model of *FvMYB82* response to low temperature and salt stress. Salt stress promotes expression of *FvMYB82*, increases the ABA biosynthesis and expression level of signal transduction genes *SnRK2.4 SnRK2.6*, and improves salt tolerance. Subsequently, ABA production stimulates the expression of *FvMYB82*. In addition, salt stress induces the expression of the crucial salt-responsive genes *AtKUP6* and *AtNCED3* and increases plant salt tolerance. Cold stress induces the expression of *FvMYB82*, and *FvMYB82* domain binds to the promoter regions of *CBF1* and *CBF2* and promotes binding of CBFs to the CRT/DRE *cis*-acting elements of downstream genes, thereby activating expression of the downstream cold-responsive genes *COR15a* and *COR78*. It also enhances plant cold tolerance.

## Data Availability

Not applicable.

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
