# Peer review of "Overexpression of a Fragaria vesca MYB Transcription Factor Gene (FvMYB82) Increases Salt and Cold Tolerance in Arabidopsis thaliana"

_ijms, 2022, doi:10.3390/ijms231810538_

Round 1

Reviewer 1 Report

Congratulations to the authors. The paper looks pretty good.

I have some suggestions to make though.

Why only low temperature and salt stress was used for experiments?

Line 60 Capital letters in between sentences.

Please justify why was this plant considered for the isolation of this gene?

You have done too much of tool based study of the protein, but no where discussed in the discussion section. In that case fig 1 and 2 can go in the supplementary as these were just extra information.

The protein was predicted to be hydrophilic, what can be the purpose of this nature justify.

Is there a better method to show the protein domain structure. Can the authors use the tools to find the domain region and present it schematically? It can be also aligned with other domains. Rather Fig 2b can be used here in place of Fig 1.

The alignment looks quite overwhelming. Better alignments can be made using tools as ESPIRIT. The resolution of the phylogenetic tree is quite bad. Can it be made in some better software? iTOL can be used after MEGA.

You can use Motif highlighter to show the important residues of the domains.

What was the germination efficiency of these transgenic seeds?

Why these gene is expressed in the immature leaves more but less in mature ones. Such statements need some justifications.

For salt stress measurement, electrolytic leakage measurement is very crucial. Can the authors generate this data?

Check the localization signal of this protein and include that study in the discussion to justify its localization in the nucleus.

How does the REDOX enzymes which were tested be fitted in the model the authors proposed?

Reviewer 2 Report

The reviewed manuscript by Li et al., Overexpression of a Fragaria vesca MYB Transcription Factor Gene (FvMYB82) Increases Salt and Cold Tolerance in Arabidopsis thaliana” is an interesting study and the authors have collected a unique dataset in their experiment. Authors reported the great role of the FvMYB82 transcription factor in response tol cold and salt stress tolerance.

However, I have general comments following;

a)     Some diverse keywords should be used.

b)     Introduction part needs greatly revise, since MYB transcription factors comprise a large and multifunctional superfamily, so authors should add information about which group of MYB family members has a great role in response to plant abiotic stress. There is no aim and objective at the end of the introduction and that should be clearly revised.

c)     There are many dissimilarities in the results for example, the authors did the analysis for physiological stress indicators including MDA or proline and enzymatic analysis, and presented the results, but are there methods for how they did it?

d)     Please clear statistical analysis

e)     Conclusion is not enough, describe a bit more

The article should be revised in most of the part including the points indicated. 

Round 2

Reviewer 2 Report

The authors revised the article critically considering reviewers' suggestions and directions. The present format can be accepted for publication in IJMS.